# Exploring Video Conferencing for Doctor Appointments in the Home: A Scenario-Based Approach from Patients' Perspectives

Dongqi Han*  Yasamin Heshmat†  Carman Neustaedter‡

School of Interactive Arts and Technology, Simon Fraser University

## ABSTRACT

We are beginning to see changes to health care systems where patients are now able to visit their doctor using video conferencing appointments. Yet we know little of how such systems should be designed to meet patients' needs. We used a scenario-based design method with video prototyping and conducted patient-centered contextual interviews with people to learn about their reactions to futuristic video-based appointments. Results show that video-based appointments differ from face-to-face consultations in terms of accessibility, relationship building, camera work, and privacy issues. These results illustrate design challenges for video calling systems that can support video-based appointments between doctors and patients with an emphasis on providing adequate camera control, support for showing empathy, and mitigating privacy concerns.

**Keywords**: Mobile video communication, doctor appointments, domestic settings, computer-mediated communication.

**Index Terms**: Human-centered computing—Empirical studies in HCI

## 1 INTRODUCTION

Telemedicine involves the use of video conferencing systems to support remote consultations with patients. Telemedicine systems can be valuable as people who live far away from medical resources or face health challenges (e.g., chronic illness, mobility issues) may find it very hard or even impossible to see a doctor in person [1]–[3]. People are now able to have video-based appointments with general practitioners using commercially available technologies like Skype, FaceTime or specialized telemedicine video systems [4]–[6]. For example, we are now seeing a proliferation of apps for video-based doctor appointments, such as MDLive and Babylon. With this comes a strong need to ensure video conferencing systems are designed appropriately in order to meet the needs of both patients and doctors.

Historically, telemedicine systems have been studied with a strong focus on specialist appointments, for example, certain chronic diseases [7]–[9] or surgery [10], [11]. In contrast, there has been less focus on system designs for patient visits with general practitioners and even less focus on understanding the design needs of patients for such systems. For example, studies have explored people's level of satisfaction and convenience with remote doctor appointments [12], [13], rather than explorations of the socio-technical challenges involved in video-based appointments and the design challenges that exist for video conferencing systems aimed at supporting appointments. This makes it unclear how to design systems that move past basic video chat software (e.g., Skype, FaceTime) capabilities.

For these reasons, our work explores in-home video appointments between people and their family physician. We were interested in understanding how patients would react to appointments focused on a range of topics from common colds to privacy invasive situations, where one could use a mobile phone and video chat software (e.g., Skype) to meet with their doctor from home. Our overarching goal was to understand what design needs and opportunities exist for video conferencing systems focused on home-based doctor appointments to meet the needs of patients, though clearly future work is needed from the perspective of doctors. We also focused specifically on conducting our study in a manner that did not expose patients directly to privacy-invasive appointments. Here we relied on scenario-based design methods [14], [15] that allow participants to examine interactions with future technologies in a grounded way.

We conducted an exploratory study with twenty-two participants who have visited doctors for general medical conditions. We were purposely broad with our sample and included diverse age groups, occupations, cultural and ethnic backgrounds. The goal was to raise as many design challenges and opportunities as possible, which comes from sampling a broad spectrum of participants. Future work should consider narrowing in on particular populations and types of visits, informed by our work that helps point to cases and situations that would be useful to explore further. We first interviewed participants about their past in-person experiences. This allowed us to learn where challenges exist, and help inform our understanding of patient needs for video-based appointments. We then used six video scenarios depicting video appointments to conduct focused interview conversations with our participants. The videos ranged from non-invasive situations such as a cold to privacy intrusive cases such as a physical exam of one's private parts. In contrast to other study approaches where we may have investigated *actual* video-based appointments or role-plays, the scenarios allowed us to gauge participants' reactions to privacy sensitive situations without putting them directly in harm's way and risking their own privacy.

Our results show that video-mediated appointments could raise issues around accessibility, relationship building, camera work to capture visuals of one's body, and privacy concerns about private information disclosure. Thus, while video-based appointments could be valuable for patients, systems to support them must be carefully designed to address these concerns. Existing commercial video conferencing systems (e.g., Skype, FaceTime) are not mapped well to the needs of patients for video-based appointments and more nuanced designs are required.

* email: dongqih@sfu.ca
† email: yheshmat@sfu.ca
‡ email: carman@sfu.ca

Graphics Interface Conference 2020
28-29 May

## 2 RELATED WORK

### 2.1 Medical Healthcare over Distance

Telemedicine systems were created to help remote populations with limited medical resources connect with physicians and specialists in urban centers [16]–[18]. They have also been designed to support people who are unable to visit a doctor in person due to difficulties such as age, disability, or diseases [3]. Doctors have been able to communicate with patients via text message [19], [20], phone call [9], or video call [21]–[23]. Telemedicine uses have also advanced over the years to serve a broader spectrum of users and not just those in rural areas with mobility issues [24], [25]. This has allowed doctors to provide more attention to patients over relatively long periods of time [10], [26] such as patients with chronic diseases [9], [27], [28].

In addition to telemedicine systems, ubiquitous monitoring instruments have been designed and deployed in home environments to aid health care [29]–[31]. Sensors have been embedded into furniture such as beds [32] and couches [33], or attached to the human body [7], [34], [35] to monitor physiological signals. Traditional diagnosis or treatment procedures become different when direct physical contact is unavailable [36]. For example, physical interaction systems can be used to transfer haptic feedback between physicians and patients [37]. Computer-aided virtual guidance has been applied to help patients conduct physiotherapy exercises [38], [39]. Factors such as system usefulness and ease of use, policy and management support, and patients' relationships with health providers have been found to be key to telemedicine system success and acceptance [40]–[42]. Security and privacy concerns have also been explored in relation to telemedicine, considering the confidentiality of medical information [43], [44]. Researchers have tried to resolve security concerns by strengthening access control [45]–[47].

Most closely related to our work, researchers have explored video-based doctor appointments through questionnaires and interviews where respondents have provided their general reactions to the idea of having a video-based appointment. From this work, we know that people feel video visits will lessen travel time and costs [48] and like the idea of having an appointment from the comfort of their home [49], [50]. Several researchers have also studied actual video-based doctor appointments. Powell et al. [13] interviewed patients after having a video-based appointment in a medical clinic office. Users reported video being convenient and only having minor privacy concerns with people overhearing the call [13]. Dixon and Stahl [12] rated patients' experiences using a video visit compared to an in-person visit after having one of both in a clinic. People preferred in-person visits but were generally satisfied with video visits [12]. In all cases, appointments were related to fairly mundane topics and privacy sensitive situations were not explored.

We build on these studies by exploring why people have specific technology preferences and social needs along with descriptions of the concerns people have with video appointments. This helps inform user interface and system design. Our work also differs in that we explore a range of appointment scenarios, some with potentially large privacy risks, which are not easy to explore with real appointments given ethical concerns. In addition, our work studies in-home usage rather than video conferencing usage in a clinic or doctor's office; this contrasts prior work [12], [13]. Usage in a home may potentially see different concerns and reactions because users are giving the doctor visual access into their home and are without medical instruments or assistance, as opposed to a doctor's office.

### 2.2 Video Communications

Video conferencing has been widely used amongst family and friends and in work and educational contexts [51]–[56]. People share views or activities via video calls, which can help create stronger feelings of connection over distance and a greater sense of awareness of others [57]–[61]. Applications range from supporting casual conversation to formal meetings [56], [58], [62]. Despite the benefits of video communication, it can still be difficult to generate the same feelings and situations via video calls as found in face-to-face communication [51]. First, people can feel that there is a barrier when watching via a computer [55], [63]. Factors such as narrow fields of view and a lack of mobility can cause users to be aware of the distance between people in video calls [55], [64]. It can also be difficult to maintain eye contact because of displacements between cameras and the video view of the remote user [40]. There are also issues with feeling like one has to continually show their face on the video call [65].

Some researchers have explored ways to increase feelings of connection over distance. For example, this has involved presenting a larger camera view and additional camera control to improve engagement with remote scenes [66], deploying interactions to support virtual shared activities over distance [59], [67], or sharing first-person views to enhance feelings of co-presence [68]. When mobile phones are used for video conferencing, one of the main challenges is 'camera work,' the continual reorienting of the camera by moving one's smartphone in order to ensure the remote person has a good view [62], [69], [70]. Local users streaming the video via their phones desire hands-free cameras that are easy to move [71]. Remote users desire the ability to gesture at things in the scene [69]. We explore the camera work needed for home-based video appointments with doctors, which has not been explored in prior studies.

We also know that video conferencing systems have been fraught with privacy concerns, despite their benefits. Online privacy issues typically relate to how users' information is mediated by media [72]. Privacy theory in video communication deconstructs privacy into three inter-related aspects: solitude, confidentiality and autonomy [73]. Solitude relates to having control over one's availability (e.g., can a person gain enough time on their own?) [73]. Confidentiality concerns how information is disclosed to others (e.g., is any sensitive background material shown on camera?) [73]. Lastly, autonomy pertains to having control over how one can interact in a video-mediated communication system (e.g., can a person choose when to use various features and for what reasons?) [73]. For example, with video calling, it can be easy to stand out of the camera's view yet still possible to see what is on the video screen or overhear the video call's audio [74], [75]. Situations like these infringe on people's confidentiality and autonomy at the same time. Across the literature, privacy concerns with video-mediated communication systems often relate to issues around showing the background of one's environment (e.g., a messy room) or a person's appearance not looking good on camera [53], [59], [74], [75]. Privacy challenges in relation to solitude, confidentiality, autonomy have not been thoroughly explored for video-based primary care appointments in the home. Our study builds on past research that explores privacy in work and family communication situations while using video communication systems.

## 3 EXPLORATORY STUDY METHOD

We conducted an exploratory study to understand what aspects of appointments patients feel are important and what benefits or challenges exist for video-based doctor appointments from one's home. Our study was approved by our university research ethics board and we took great care and caution to conduct our study in a manner that did not increase privacy risk for patients.

### 3.1 Participants

The study enrolled a total of 22 participants (17 females, 5 males) who had visited doctors. We recruited participants through snowball sampling, posting advertisements on social networks and university mailing lists. The gender imbalance was unintentional and based solely on who responded to our participant call and was willing to participate. Seventeen interviews were done in person either on our university campus or at participants' homes, whichever they felt comfortable with. Five of the interviews were done over Skype. The participants were all adults within the age range of 19-71 (average=37, SD=16). Participants had a range of cultural and ethnic backgrounds, including individuals with European, Asian, and Middle Eastern descent. To reach a diverse data set, we recruited participants from different age groups, occupations, cultural and ethnic backgrounds. We purposely chose a diverse group of participants so that we could find out as many benefits and challenges as possible in terms of designing technology for supporting video-based appointments. Thus, the goal was to be exploratory such that future studies could then investigate the areas of opportunity and concern revealed by our study in more detail. Some of the participants visited the doctor regularly for conditions such as blood pressure check, gout, anxiety control, arthritis, depression, digestive system issues, etc. Others visited the doctor only when sick or for checkups.

### 3.2 Method

We used semi-structured interviews to gain an in-depth understanding of patients' experiences with in-person doctor appointments and thoughts about video-based appointments. Each interview contained two sections. In the first section, participants talked about their past doctor appointment experiences. In the second section, they were shown six video scenarios, each of which dealt with a distinct medical condition, and we interviewed them about their reactions. Participants could choose between a female or male interviewer in order to feel more comfortable sharing their private medical experiences or their personal opinions. Eleven participants (7 females, 4 males) had an interviewer of the same gender; 10 female participants had a male interviewer and 1 male participant had a female interviewer. The interviews lasted between 50 and 90 minutes.

#### 3.2.1 In-Person Experiences

To learn more about patients' face-to-face appointments with doctors, we asked them to talk about their past appointments that they felt went well or not. The goal was to use this knowledge to understand what aspects of in-person appointments should be maintained or improved in a video-based appointment. Furthermore, the shift of doctor appointments from in-person to online might bring both challenges and opportunities that were unknown to us. Thus, an understanding of in-person doctor appointments would benefit our thoughts on how to design video conferencing systems. It would also act as a form of baseline.

We purposely ground this interview phase in questions about specific appointments, as opposed to more general thoughts, to acquire detailed and specific data. When recalling these visits, participants were asked to describe the details, such as their conditions, the examinations performed, how the diagnoses were made, what treatments were provided, whether they had follow-up visits, etc. In this way, the questions would help them recall as much information as possible about the appointment. As examples, we asked, "Can you tell me about a visit you felt that went very well (or not well)?" and "How did you describe the situation to your doctor?", "What worked well?", "What did not work well about the visit?" At the end of this interview section, participants were asked about their general opinions on the necessity of face-to-face office visits (as opposed to video-based appointments) and what factors they believed were important during the appointments. This section of the interview lasted 20 to 30 minutes.

#### 3.2.2 Future Scenarios

##### 3.2.2.1 Scenario Planning and Production

Next, we conducted scenario-based interviews with participants. This method was selected based on a lot of careful thought and planning. We wanted participants to understand the concept of video conferencing between a doctor and a patient and ask them about specific attributes of such appointments, which may be hard to imagine. Yet we were cautious that we did not want to infringe on the privacy of our participants as we wanted to explore topics that were both commonplace as well as privacy intrusive. For these reasons, we employed an approach similar to scenario-based design [14], [15], which is often used in the early stages of design cycles in the field of human-computer interaction. The goal is to elucidate the use of novel technologies that might not exist yet or be widely used. This approach is able to engage users in exploring both the design opportunities and challenges that might exist for a technology through storytelling and conversation. With this type of method, participants can be shown pre-recorded videos of people and design artifacts, which are then used as a conversation piece to discuss future technology usage. In our case, we wanted to illustrate aspects such as what could be seen on video during an appointment, including the environment, facial expressions, gestures, and one's body, and how smartphones might need to be oriented or used to capture such information.

Other study options might involve investigating real video-based appointments or letting participants role-play as opposed to showing them videos; however, we felt there were serious ethical challenges. First, it would be difficult and highly privacy intrusive to observe real appointments about privacy-sensitive topics, e.g., talking about drug usage or domestic abuse, conducting a visual exam of one's private areas. Second, role-playing such appointments could similarly be awkward and privacy intrusive. In contrast, we felt that pre-recorded video scenarios would allow us to gauge participants' reactions to privacy sensitive situations without putting them directly in harm's way and risking their own privacy. Pre-recorded videos would also allow us to have control over what participants saw, as they would each see the same situation. This would mean we could learn about everyone's reactions to the same situations, and we could explore multiple appointment topics with each participant rather than just one.

Prior to the study, we planned and pre-recorded six sample doctor-patient appointments using a video conferencing system. This involved brainstorming possible appointments and the likely benefits and challenges that might exist for patients and doctors. We narrowed down a large list of scenarios to a set of six that we felt mapped to a range of experiences. We then iteratively generated scripts and storyboards for each video. These were reviewed with a doctor who conducted video-based appointments to ensure the appointments we depicted were realistic.

We chose scenarios based on several aspects. First, we wanted the scenarios to cover common medical conditions where appointments would normally be conducted in a clinic, including conversation between the patient and doctor, visual examinations, or physical touching. Second, we selected scenarios that would require a variety of camera work to facilitate the video call, e.g., orienting the camera to have a view of the patient's whole body, face, or particular areas like the mouth. Third, we selected scenarios with different levels of potential privacy concerns to receive a variety of reactions from participants. Some appointments were felt to be somewhat mundane and non-problematic (e.g., a cold), while others were purposely meant to

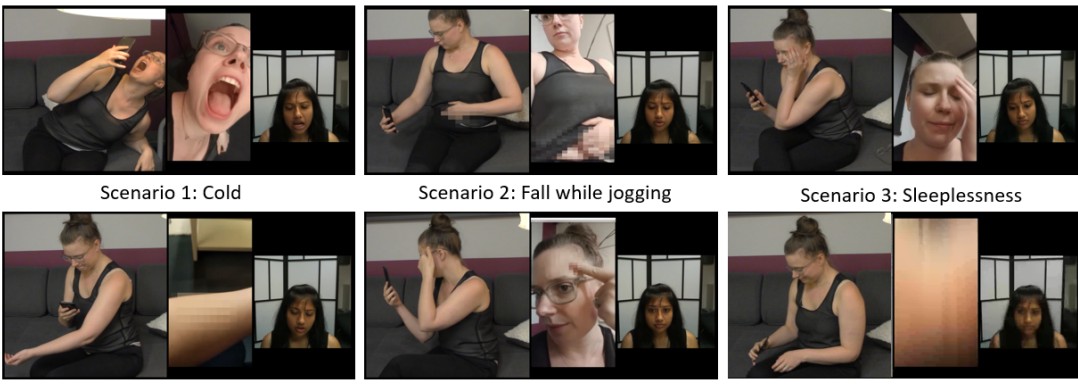

Figure 1: Images depicting the video scenarios that were shown to participants. In each video, from left to right: third-person view of patient, camera view of patient, camera view of doctor.

offer problematic situations in different ways (e.g., problems with conversations, problems with what is shown on camera). The resulting scenarios were:

**1) Cold:** The patient had a cold and sore throat. The doctor asked the patient to explain the symptoms and show their throat with the mobile phone camera.

**2) Fall while jogging:** The patient described falling down while jogging and was asked to show the injuries. The patient followed the instructions of the doctor to uncover their stomach region and press on different locations to inspect for internal injuries.

**3) Sleeplessness:** The patient explained that they were having sleepless nights. They were asked about alcohol and coffee intake. The doctor said they would send a referral to a counseling office.

**4) Drugs:** The patient had a rash on their arm. After excluding the common possible causes, the doctor asked the patient about using drugs which made the patient feel awkward as they didn't realize the possible connection.

**5) Domestic abuse:** The patient described being dizzy and having a bruise on their forehead. They were asked questions about their cognitive competence, then the patient confided in the doctor that there was partner abuse.

**6) Private parts:** The patient and doctor discussed results from an annual physical exam. The doctor asked about the patient's sexual history and a lump on the patient's groin area. The doctor instructed the patient to show their private parts and the doctor performed a visual examination.

Next, we recorded each video scenario twice within a home setting in our lab, once with male actors for both the patient and doctor, and once with female actresses for both patient and doctor. The videos used the same basic script. Figure 1 shows an image from the female versions of the videos. In each video, participants were shown: 1) a third-person view of the patient holding their mobile phone so they could understand the camera work that was needed to capture the video (left side of the video); 2) the camera view of the patient (the middle of the video); and, 3) the camera view of the doctor (right side of the video). When the videos revealed the patient's body, we blurred parts that might normally be hidden under clothes. This was to protect the privacy of the actors in the videos. For Scene 6, we did not record the portion of the video showing private areas. Instead, we showed a masked portion of video. Videos were between 1.5 and 2 minutes each.

#### 3.2.2.2 Scenario-Based Interview Method

In the study, we showed and asked participants questions about the six scenarios, one at a time. Participants watched the videos that mapped to their gender selection. We felt that this mapping might generate stronger empathy from our participants and help them to imagine how they would feel if they were in the same situation as the actor. We did not counterbalance the ordering of the scenes as we wanted to ease the participants into the idea of video appointments with somewhat mundane situations first and our work was meant to be exploratory rather than a carefully controlled experiment. This does have the limitation that the order of the scenes could have affected participants' thoughts about them.

After watching a video, participants were asked to provide reactions to the specific situation, where we asked what they saw as the benefits or challenges of using a video call for the appointment. These questions included, for example, "How would you feel if you were the patient in the video?", "How would you compare an in-person appointment with that in the video call?" We then repeated this for each video, one-by-one. We also asked for their opinions on camera control, privacy concerns, and the use of different types of technologies. The scenario-based interview lasted about 40 minutes. Participants received $20 for participating.

### 3.3 Data Collection and Analysis

All the interviews were audio-recorded and fully transcribed. We recorded videos of the participants watching the scenarios and talking about them (with permission). Two researchers analyzed and coded the data. Each researcher coded the data separately, then one researcher merged the coding and conducted additional analysis. The researchers also discussed their analysis and coding. We used open coding to label all the findings in the transcripts. Afterwards, through the use of axial coding, we formed categories such as privacy, benefits and challenges of using video calls, trust, physical examination, etc. Lastly, selective coding was used to find high-level themes including *accessibility*, *empathy and trust*, *camera work* and *privacy concerns*. In the following sections, we describe our findings under the four main themes. Quotes from participants are presented with participant ID as P#. We intermix descriptions of patients' past experiences and their opinions of video appointments as a way to further analyze video appointments.

### 4 ACCESSIBILITY

Participants felt that video appointments could create a lower barrier for accessing one's doctor than in-person appointments. Participants described visiting their doctor based on their own judgements around when it was important to do so. Many of them

said they would not bother to see a doctor for what they felt were minor things (e.g., a general cold or bruises) and perform an analysis by themselves, sometimes with the aid of web searches. Participants said that often they were not sure whether they should visit a doctor or not. Some felt that a lower barrier to meeting with one's doctor might make it easier for them to meet about more situations where they were unsure as to whether an appointment was necessary.

*Instead of you waiting for a week to visit the doctor you can use this system to have primary comfort to know how serious or not the problem is until you find an appointment time. -P11, female, 33*

## 5 EMPATHY AND TRUST

Relationship building is one of the essential aspects of doctor-patient communication. Similar to prior research [76], participants told us that body language was important during conversations with their doctor when in-person. Conversations involved eye contact and body gestures. By looking patients in their eyes, nodding while listening, or using hand gestures when explaining things to them, the doctor could let patients feel like their conditions were heard, their feelings were understood, and their problems were trying to be solved.

Participants felt that a video-based appointment would cause changes to the ways that body language was conveyed and seen. For example, doctors could be multi-tasking on their computer or not fully paying attention.

*I think over a video call it's hard to know if the person's attention is only on you because they might have other tabs open and stuff…Whereas if you're in-person, you know through their body language and through their eye contact that they're actually focusing on you. – P1, female, 19*

The user interface in our video scenarios tended to only show the doctor's face and shoulders, akin to a typical video chat (Skype) call. Yet participants described wanting to see more parts of the doctor's body during appointments. For example, one participant wanted to see the doctor's face and upper body, including their arms and hands in case the doctor gestured with them. This would make the appointment feel 'more real'. One participant said it might be difficult to have eye contact with the doctor if the camera was not at the right angle.

## 6 CAMERA WORK

We talked with participants about the camera work that would be needed within video-based appointments and they saw various aspects of it in the video scenarios. By camera work we refer to the orienting of the smartphone camera such that it can capture the information desired by doctors.

### 6.1 Visual Examinations

First, all participants talked about the doctor doing visual examinations of their body or parts of it during their previous appointments. When it came to video-based appointments, participants expressed concerns about whether the camera could clearly show body parts in order to support visual examinations by the doctor. There were several conditions in our video scenarios that contained visual exams, e.g., showing down one's throat, wounded legs and foreheads, rashes, and genitals. In these cases, participants felt that the camera resolution, color accuracy, network quality, and light intensity could be a problem. Participants felt that visual checks would be less accurate over a video call than in-person. This could cause them to lose some trust when it came to their diagnosis. Based on their past video chatting experiences, several participants noted that the quality of static images was much better than that of showing via video, as video resolution is highly limited by network bandwidth. Thus, they felt that images may be better for information sharing in some situations. One caveat is that this could require careful camera work in order to hold the phone steady for a picture.

*I think if there's a camera that can just take a snapshot of your throat…just like when you go and get your x-ray of your mouth for your teeth…which could automatically be sent to the doctor rather than you go "aww". – P13, female, 34*

We asked participants if they would have different reactions to aspects of camera work if they were using a desktop computer with a webcam or a 360-degree camera that could automatically capture the entire scene. In this case, the doctor could look at various parts of the patient's body without the patient having to move the camera around. Participants generally felt that the extra wide field of view provided by a 360-degree camera would not aid visual examinations. Participants felt that mobile phones were better for situations when they wanted to show body parts as their phone was highly mobile and they could bring it close to their body. However, one participant said a mobile phone camera would be inconvenient when they needed to perform certain actions with two hands, such as lifting their shirt and pressing their abdomen at the same time (e.g., *fall while jogging* scenario). She also pointed out that it would be tiring to hold their phone all the time when talking with the doctor.

*I can see that, given a long consultation, the patient probably gets tired that she has to hold the phone and it's not comfortable anyway…The patient only has two hands to set and hit the body. With the mobile phone, she really needs one hand. – P17, female, 42*

Lastly, participants talked about the importance of the doctor seeing everything that occurred in a doctor's office. This included the way that the patient was sitting in an office chair to how they moved to an examination table.

*The physical examination of the patients starts when the patient opens the door…you take a look at their appearance, the way that they walk, if they are so tired, how they carry their bodies, how they walk. The general appearance of a patient is so helpful. When you are Skyping with someone or you are Face Timing with someone, it's really impossible to get the general idea of how the patient is walking, how the patient is doing stuff. – P8, female, 31*

Participants felt that every subtle detail was important for the doctor to see because it could relate to things that the patient did not think to tell the doctor. For example, one might not think to tell the doctor that their foot was sore after a bicycle fall but this could be noticed when a person walked. Participants noted that such aspects might not be visible during a video-based appointment since a mobile phone's camera would likely be pointed at the patient's face rather than their entire body. The room's lighting or Internet bandwidth may also compromise what was visible.

*You can find so many precious points about so much precious information about the patients by doing physical examination. For example, sometimes patients forget. You are doing the physical examination on their chest and you see a scar on their sternum, and you ask them what this scar is, and the patient is like, 'Oh, now I remember. I had a surgery 20 years ago or something. I forgot doctor. Sorry.' – P8, female, 31*

Participants also talked about the chance of patients lying or hiding details from their doctor. While nobody admitted to doing so, participants felt it would be easier to lie in a video-based appointment. Supposed indicators of lying, such as subtle eye movements or discomfort while sitting, may be harder to notice. Some commented that people may also be more inclined to lie over a video call because they were 'online' and not in-person. These issues were seen as compromising a doctor's diagnosis.

## 6.2 Physical Touch

Participants' past in-person appointments often involved palpation to feel their body. This was directly explored in the *falling while jogging* scenario where the patient in the video had to press their own abdomen following the doctor's instructions. Some participants believed that the patient could perform this action on their own as long as the doctor could clearly see the patient's actions. The challenge, as previously mentioned, was that this could require very careful camera work in order to both capture the action on video and touch oneself at the same time. Participants also talked about not being properly trained in some cases and having a hard time following a doctor's instructions when it came to physical touches; thus, people were apprehensive about doing this work as the patient.

*The patient could probably apply less pressure than needed to feel versus a doctor. A doctor can physically tell if it's serious or not instead of having patients to let him know. – P6, male, 24*

## 7 PRIVACY CONCERNS

Participants had several privacy concerns when it came to video-based appointments like those in our scenarios, including issues with both video and audio.

### 7.1 Private Visuals

First, we talked with participants about their reactions to showing visuals of themselves on camera that might be considered privacy sensitive. All participants were fine with showing non-private areas of their body to remote doctors, as they saw in our scenarios. Yet when it came to show private body parts (e.g., groin area, chest) over video, all participants showed concerns about privacy, in particular their confidentiality and autonomy, and preferred to visit their doctor in-person. Many participants thought it was weird to show their private parts over a video call as they had never experienced it before. They were concerned about the security of the video link and worried that it may get 'hijacked'—again, issues around confidentiality. Several participants also said that they did not know if anybody was in the doctor's office but off-camera and able to see or hear the appointment. Thus, they had concerns in relation to autonomy and their ability to participate in the video-mediated space in a way that they desired. In contrast, they felt that when in a doctor's office in-person, the patient would know for sure who was in the room because they could see all areas within it.

Participants also had privacy concerns when it came to situations such as the *domestic abuse* scenario and raised several specific issues, albeit these varied across groups of participants. Participants talked about what it would be like to be in a situation involving domestic abuse. Several participants said that staying at home and having a video appointment was a better choice as the private information, bruises in this case, would not be visible to people other than the doctor. They thought that as a patient they might feel uncomfortable outside their house and be noticed by people on their way to the doctor's office or in the waiting room. Other participants talked about how a video appointment at home could present additional risk since an abusive partner could come home unexpectedly. They felt that when in-person, only the patient and doctor would be in the doctor's office. The patient would be safe, and the conversation would be private as well.

*Because you don't want neighbors to see anything or a random stranger to think, 'Oh my god, she got beat up. She's in a bad situation.' And in the conferencing, she could just talk more openly and say, 'Okay, I'm sharing this with you. You're the only one that sees it.'. – P21, female, 68*

*Consulting with a doctor at home will increase the risk of abuse again. – P3, female, 21*

Given the privacy concerns that participants expressed, we asked them about possible ways of mitigating their concerns. For example, we talked with them about the possibility of blurring their face in the video feed during situations such as the *private parts* and *domestic abuse* scenarios so they would feel more comfortable with the appointments and have a video call in more of an anonymous fashion. This was seen as being valuable by eight participants though two of the remaining participants pointed out that it could make it harder to get accurate diagnoses since the doctor would not know the patient's history. Doctors may also not be able to understand the patient's facial expression, which could help them assess the severity of a situation.

*I think [blurring faces] is very good. For example, when you want to go there and talk about drinking or marijuana or private parts, these kinds of things. I know people that don't go to doctor at all just because they don't want to talk about it with another person. – P12, female, 33*

*You can read the expressions of the people's face, eyes. 'Okay, this lady is really scared … or she knows it's a minor thing, so she's not really worried about it'. – P21, female, 68*

Participants were asked if they would feel any different in terms of privacy if the doctor was a different gender than they were. Five female participants explained that they preferred a doctor of the same gender for health problems that they felt were private. All male participants felt okay with doctors of both genders regardless of the situation.

*Especially if it's not my regular family doctor, I would not want a male there. Actually, even if it was my family doctor, I usually try to find the public nurses, like female. – P9, female, 32*

*I guess I don't really care what gender my doctor is, as long as they're professional. – P7, male, 27*

### 7.2 Private Conversations

Second, we talked with participants about the kinds of conversations that were occurring over the video appointment scenarios and how comfortable they would be in having such conversations with a remote doctor and responses varied. Some participants believed that telling the doctor about their medical conditions was not embarrassing as the doctor was professional and they should be honest with them and explain everything. In contrast, other participants felt embarrassed about the conversations in all of the scenarios except *cold* and *fall while jogging*. Most female participants felt embarrassed talking about sensitive issues such as relationships, sexual practices, abuse, and drug consumption. None of the male participants had the same concerns. Participants also commented that they felt some people may be less inclined to have conversations about sensitive topics due to cultural backgrounds and taboo topics.

*People have such different cultural backgrounds that something that is not taboo with some person could be really taboo to another, and really affecting their ability to communicate what's really going on. – P16, female, 38*

### 7.3 Camera Control

Third, we probed participants about privacy in relation to video capture and who had control of the camera, them or the doctor. We asked participants whether they would be okay giving up control of the camera to the remote doctor, if it was possible. For example, one could imagine placing their phone in a stand that had remote controlled pan and tilt features. In general, the responses were based on the amount of trust that a person had built up with their doctor and how strong they felt their relationship was with this person. This was the case for the majority of the participants who said that giving up control of the camera to their doctor was fine because they trusted their doctor

and felt that if the doctor had control of the camera that they would be able to more easily acquire their preferred view point.

*He would know exactly what he's looking for. Or he'd be able to focus it better to take a look at what it is that he needs to look at or tell you exactly where to press to figure out what is the extent of the injury or just so that he has enough information to make the correct diagnosis. – P20, female, 61*

A few participants said they would only give up control if it was necessary. One wanted to be informed before and during the call about what the doctor wanted to see. P11 likened this to an experience she had where she needed online services to help her fix her computer. She felt that having patients be able to monitor what the remote person was doing with the camera would dispel privacy concerns.

*[Online support] always asked me if they can control my computer…The first time I did it, well that's a pretty big step for me. But then I realized I can see exactly what they're doing and then they're working in an office space. I think I can trust them. – P1, female, 19*

One participant talked about giving the doctor more control, such as the ability to capture images and draw annotations on them. This could help illustrate things to patients.

*Maybe if the doctor could take screenshots and then annotate them, and then show those to the patient, saying like 'Oh, you need to take care of this part of your mouth, like this tooth,' …Or 'Oh, I see this here,' and then they circle it. 'Can you apply this kind of medicine to that part of your mouth?' – P7, male, 27*

We also asked participants about newer technologies that might be used as a part of video appointments to give the doctor a better view of the patient or their environment. For example, we asked about 360-degree and wide field of view cameras. Some participants said they felt it was unnecessary for a doctor to see an entire room. Others were okay, again, if they knew what a doctor was looking at and if it was useful for the appointment and a diagnosis.

## 7.4 Video Recording

Participants talked about the possibility of the video calls being recorded and this was troubling. Ten of them expressed concerns that they did not want their video-based appointments to be recorded. Moreover, some expressed concerns that the doctor might capture screenshots of the video without their knowledge or permission. For example, one participant talked about the potential that exists when people have access to private information:

*I worked in a computer networking in [organization name], we could access passwords of the users, but we were not allowed to tell this to users. We never used it. But we could have. – P12, female, 33*

Two participants said that it could be valuable to have video recordings of appointments in order to have a more complete history of one's medical record, yet there were large concerns over who would have access to this video data.

## 8 DISCUSSION

We now discuss our method and results to explore the challenges and design possibilities for video-based appointments between doctors and patients.

## 8.1 Scenario-Based Design

We employed a study method that built upon scenario-based design given difficulties and privacy risks in observing and talking with participants about real appointments. By presenting participants with vivid and graphically rich video clips, we were able to illustrate a series of scenarios that we wanted to explore in detail. This was far beyond would we likely would have been able to achieve through verbal descriptions alone. Participants reacted positively to the method and were able to engage in detailed conversations with us as researchers. Thus, we feel our approach is especially helpful when the situations one wants to explore are non-existent or rare at present time. It is unlikely that they are using them for risky and privacy-sensitive situations. Of course, having participants participate in actual doctor appointments would move reactions beyond the types of speculations that participants had in our study. However, it would be critical that studies of such appointments be carefully designed to counterbalance the possible effects and ethical dilemmas found with privacy intrusive studies. We feel that one value coming out from our study is that we now have a better understanding of how people will react to privacy-intrusive video-based appointments, which can help researchers understand how to plan studies with actual appointments in a way that minimizes privacy and ethical risks.

## 8.2 Privacy Aspects in Sensitive Situations

It is clear from our results that video visits are currently not a replacement for all types of doctor-patient appointments. Participants saw video-based appointments as being supplementary to seeing their doctor in person. Some privacy invasive situations, for example, involving showing one's private areas or talking about sexual-related situations, are clearly not great candidates for video-based appointments for now. Some people are also unwilling to disclose private information, be it visually or aurally communicated, and rightfully so. Thus, they are concerned about the confidentiality of information and how it is disclosed over video [73]. Several concepts related to confidentiality were reflected in our findings, including information sensitivity, concerns about video fidelity and control over its capture [73]. First, there were issues related to impression management. Having patients reveal sensitive information could create embarrassment or identity issues with self-esteem. Some of our participants felt it would be valuable to blur their faces, which would allow them to 'detach' themselves from their identity and make the conversation less personal. Second, fidelity involves the persistency of information [73]. In the case of video-based appointments, this could include the recording of conversations. Future design work could explore how to reduce patient anxiety about the possibility of video recording. Third, control involves knowing that video is only being seen and heard by the doctor and patient, and no other parties [73]. In face-to-face appointments, patients are able to know who is in the physical space of the office. This can become difficult to know over video. It reflects a common issue that has resonated in the video communication literature around people being disembodied in a video-mediated environment (being able to see / hear while off-camera) [75]. This suggests the design of cameras with larger fields of view for the doctor's office so that patients can maintain an awareness of the space. Our results also reveal that there could be some situations, even those involving very private information (e.g., conversations about domestic abuse), where some people may feel more comfortable with a video appointment compared to an in-person one so that they can do so from a location of their choosing. This may make the appointment more comfortable for them and avoid being exposed to others outside of their home. In this way, the video appointment could help control the confidentiality of sensitive information. We caution though that none of our participants reported having experienced domestic abuse, so these findings should be validated through further study.

## 8.3 Visuals of the Patient and Doctor

Patients saw value in having the doctor see their entire body and area around them rather than just their face in case there were things that the doctor might notice what they didn't think to talk about or explain. Yet there were hesitations around cameras that might have wider fields of view, such as 360-degree cameras, because of what else might be captured by the camera in their home. This suggests design explorations into methods that might provide doctors with broader views while still balancing the privacy concerns of patients. For example, one might imagine systems that allow patients to selectively blur or replace the background [77] in video feeds that they feel are private, but this would need to be done in a way that still allows doctors to understand what is happening in the video for proper diagnosis. Patients were generally fine giving up control of the camera to their doctor, if there was trust in the relationship and they knew what the doctor was looking at. Thus, there were few concerns with autonomy over how one participates in the video-mediated space. This illustrates the importance of doctor-patient relationships given that patients are willing to give up some autonomy based on their trust in doctors. As is conceptualized by Palen and Dourish [72], privacy involves the management of boundaries which can be dynamic according to contexts or actions. Doctor-patient relationships work as boundaries within video appointments. A well-established relationship could transfer the autonomy from the patient to the doctor's side in order to support the examination on the patient. This could factor into design solutions where a doctor may be given greater control to, for example, remotely pan a camera around the patient's environment to see them better, providing that the patient knows what the doctor is looking at.

Relationship building was considered essential in doctor-patient communication [76]. Turning to patients' views of the doctor's environment, participants wanted to see body language and ensuring eye contact with the doctor. It reflects the challenge of building rapport using non-verbal behaviors on camera. While such visuals are important in mobile video calls with family/friends [51], [55], the element of trust that they evoke with patients feels different and, in some ways, more critical in building doctor-patient relationship over video [78]. As mentioned, designs that include a camera with a larger field of view might allow one to see the whole body of the doctor, including body language, which could help build rapport with patients. However, such designs should be carefully thought through as multiple factors and challenges are intertwined.

## 8.4 Appointment Accessibility and Awareness

We note that the flexibility that might come with video appointments could easily bring caveats. For example, participants tended to feel that doctors would be more accessible to them if video appointments were available. Mobile apps which provide virtual visit services may encourage patients to meet with unknown doctors online in the present moment as opposed to waiting a few days to see their own doctors. This might cause an overutilization of video appointments and a loss in the continuity of care over time. This suggests there are design opportunities to better link medical records with apps that permit video-based appointments so that information can more easily be shared between providers.

According to our findings, patients had challenges knowing whether their situations would be appropriate for video visits. This suggests that they are design opportunities for exploring systems that may help patients screen themselves to see if a video appointment would be appropriate. This could be implemented using questionnaires that patients answer along with decision tree algorithms, or with medical professional assistance (e.g., a nurse asks screening questions over a short video call).

## 8.5 Camera Work and Visuals of the Patient

Like mobile phone research for video calls between family and friends [62], [69], [70], we, too, saw challenges around camera work and easily capturing the right information for doctors. There were pragmatic issues like camera lighting, but also issues around holding phones to show body parts while also being able to see the doctor's reactions. Compared to the related literature [51], [53], [54], the complexities around camera work seemed to be more difficult in our study situation. Mobile phone calls with family/friends tend to not involve trying to show specific body parts and instead, focus on faces or surrounding contexts [54], [58]. Video calls with doctors may try to expose highly accurate images of one's body parts such as the neck, abdomen or back. These areas can be difficult to capture using ordinary mobile phones. In addition, it can require high quality lighting and proper camera orientation. This contrasts casual conversations with family or friends. Video appointments might also require that patients perform particular camera movements that they typically do not do on a tablet, laptop, or phone when using existing video chat tools (e.g., Skype, FaceTime). Because the camera is coupled with display showing the camera's view (e.g., the phone), it can be hard to direct the camera to a particular area while also looking at the screen to see what is in view.

These challenges suggest the need for tools that make it easier for patients to perform the necessary camera work during a video appointment. Tools could focus on ways to hold and move a camera for easier capture. For example, video conferencing software may include on-screen visuals [79] to show patients where to move the camera to capture an area of one's body, or augmented visuals [80], [81] to guide patients to perform certain actions. One could imagine customized versions that help users capture areas of their body after selecting a particular body part, e.g., clicking on 'knee' in the application could trigger visuals that guide the user to capture all views of a knee.

One could also think about hardware tools or devices that would make it easier to hold a mobile phone camera or set it down in order to capture body parts that are at awkward angles or locations. People commonly use 'selfie' sticks or phone stands to capture pictures presently. One could imagine custom designed apparatus for video-based doctor appointments that let a person more easily hold or set down their mobile phone to capture a body part on camera. Designs could also explore the decoupling of the camera device from the display device. For example, it may be easier to perform camera work during a video appointment if the camera could be held in one hand to show a body part, while the user looks at a separate display to see what the camera is capturing. This is not normally done with existing video chat tools since people often use devices with cameras built into them. External cameras could be highly valuable for video appointments.

## 9 CONCLUSIONS AND FUTURE WORK

Overall, our research points to the value that video-based appointments could bring to patients, from a patient-centric perspective. We have pointed to a variety of opportunities for additional design work in order to mitigate camera challenges and privacy concerns. While we are cautious to not suggest more specific design directions and implications for design without additional design work, clearly existing video chat technologies (e.g., Skype) do not map to the specific needs of video-based appointments. Their two-way calling model would mean that patients could try to connect with the doctor at inopportune times. They are also limited when it comes to the camera work that

would be necessary and valuable for video-based appointments, including features to help mitigate privacy concerns and guide the user in capturing areas of their body on camera.

Naturally, future work should study the needs and experiences of doctors when it comes to video-based appointments. These may offer alternative needs than the patients in our study had, which might suggest further design accommodations and balances that need to be made to address the needs of both groups of users. Our study is limited in that we had a large number of female participants by chance. Few males contacted us to participate. Future work should further explore the concerns of males as well as others. We also recognize that other cultures might feel differently towards video-based appointments. The patients we studied were mostly participating in the health care system in Canada that is publicly funded where they can visit the doctor at any point in time and not have to pay for the visit. This could have affected their viewpoints in our study.

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
