# OpenReview forum: "Exploring Video Conferencing for Doctor Appointments in the Home: A Scenario-Based Approach from Patients’ Perspectives"
_graphicsinterface.org/Graphics_Interface/2020/Conference — GI 2020_

### Official Review · AnonReviewer1 · 2020-01-08
**Exploring Video Conferencing for Doctor Appointments in the Home: A Scenario-Based Approach from Patients’ Perspectives**

**Confidence:** 4
**Rating:** 9

**Review:**

The paper offers a practical account of how-to better design video conferencing platforms for patient-doctor appointments. Using a scenario-based design method and a range of user interviews, it points to important issues about how accessibility, relationality, privacy and information disclosure concerns, and humanely interaction can be incorporated into the thinking and design of telemedicine systems. A well-motivated and situated work that made emphasis on the socio-technical challenges and potential design needs and opportunities of a patient and not some abstract or idealised understanding of telemedicine systems.

Methodologically, the paper clearly outlines and provided some relevant justification for the chosen research–design approach; the sampling approach, size and characterisation; and the sensitivity towards various interactive scenarios. Generally, the study method, the method of data collection and visualisation, and the level of sensitivity and relativeness of the authors towards not only being reflexive but also practising some form of relational accountability and openness shows the originality of the methodology section. The analysis might be considered providing a thick description of what was conducted, how, why and to what extent the findings can be considered representative of the 22 participants – therefore considered a clearly detailed paper.  From my understanding, this suggests originality, not only methodologically, but of its situatedness within the context of the literature.

The few reservations I had relates to the recommendation for future work at the start of the paper, specifically ‘Future work should consider narrowing in on particular populations and types of visits’– this should have been at the end as it might tip off the focus of the narrative having such statement in the introduction. It makes me wonder why not focus on a particular population, and not broadly.  In conclusion, Dourish's 'implications for design' ought to be cited where necessary. It will boost the paper to include a few lines about the implication of the methodological sensitivity, the design of the research, and how the methodological/analytical aspect of the empirical contribution might reframe our thinking and understanding of telemedicine systems. I believe this will exemplify the significance of the work to the audience.

Generally, I believe the paper makes a significant contribution to our understanding of designing video-conference platforms (and those that are compatible with smartphones) specifically tailored to the need of patients and in consideration of important factors that commercially available systems might have taken for granted. It also points to specific challenges – issues of privacy, accessibility, visuals, camera work, trust, and relationality – and provides possible insights for designing telemedicine tools specific for a virtual appointment.

---

### Official Review · AnonReviewer2 · 2020-01-08
**Excellent study design in an important research area**

**Confidence:** 4
**Rating:** 9

**Review:**

This paper presents the results of exploratory contextual interviews around the use of video calling for health appointments with a particular focus on patients’ views on privacy issues. Using six scenarios, increasing in degree of privacy concerns, participants shared their thoughts around accessibility, privacy, and other issues. They find that, rather than relying on existing video calling tools, new systems need to be designed to consider the privacy and practical issues specific to this space.

This exploration of perspectives on video calling for medical appointments focuses on input from patients in order to present an important point of view on the issues surround this technology. The in-depth focus on patients allows for a clear understanding of their perceived benefits and issues, beyond the potential practicality, and reveals patients’ specific needs. The qualitative study design allows participants to react to the increasing privacy needs in context and based in their own experience.

The paper is overall well-written, and I have only one suggestion. While the discussion is well situated in the existing literature, having more clear design suggestions would give the paper a stronger take away. What needs to be done specifically to support this area of video calling? Are all the areas for improvement equally important or do patients see some as having higher priority? Providing a more concrete presentation of the design implications found in this study would help readers to understand the overall contribution.

Generally, this is very interesting research into a particularly sensitive area of research, and the focus on the voices of patients provides a clear view of their valid concerns and what needs to be done to support them.

---

### Official Review · AnonReviewer3 · 2020-01-09
**The paper explores an important research question. The method is appropriate and the results are well-organized.**

**Confidence:** 4
**Rating:** 8

**Review:**

The paper explores the design challenges of video conferencing for doctor appointments. This work combined semi-structured interviews with a scenario-based interview to elicit patients’ comments on their actual doctor appointments and the reactions to a set of staged video-based appointment scenarios. Interviews and comments on the scenarios were analyzed using open coding, and the results are organized into four themes.

Video-conferencing for doctor appointments could potentially be beneficial for patients who live far away from hospitals or have chronical diseases. Thus, it is important to understand its design challenges. The method is appropriate; the results are well-organized and the discussion highlights key design challenges derived from the findings. Overall, the paper is well written and easy to follow.

Although I am positive about the paper in general, I do have the following concerns:
The paper briefly talks about some limitation of the video-conferencing based doctor appointments. For example, doctors could not physically touch patients. However, the paper does not discuss the limitations of video-conferencing based doctor appointments based on the study findings and what could be done to mitigate such limitations or the design implications for such limitations.

In the discussion of Camera Work and Visuals of the Patient, the paper mostly focuses on issues with manipulating a first-person camera (e.g., mobile phone camera). However, the scenarios provided two camera views (both the first-person and the third-person). What is the camera work for operating a third-person camera? What are the design implications for the third-person camera?

Typo:
“…they are design opportunities for exploring systems…” -> “…there are…”

---

### Meta-Review · Area_Chair1 · 2020-01-12

**Recommendation:** Accept
**Confidence:** 5

**Metareview:**

Reviewers acknowledged that the paper is interesting, well-written, well-motivated, and addresses an important problem in an area that is relevant to the HCI community. It is also methodologically sound. They also highlighted the thoroughness of the methodology employed.

However, they also highlighted some issues with the paper that the authors need to address: Below, I summarize the key issues. However, I encourage the authors to read through the reviews carefully and address all issues highlighted by individual reviewers:

-	Missing citations and some reflections on implications of the methodological sensitivity for future designers [R1].
-	Some misplaced sections, e.g., some future work included at the beginning instead of at the future work section, it breaks the flow [R1].
-	Lack of clear and concrete design suggestions for future research in the area of video conferencing design and related areas [R2].
-	Lack of discussion of the limitations of video-conferencing based doctor appointments especially the third-person camera and how to overcome them [R3].

Despite the shortcoming and highlighted weaknesses, the reviewers believe the paper hold some potential. I also believe that, although the issues highlighted are very important and must be addressed in the final version, they do not require significant changes to the paper. Hence, I recommend that the paper be accepted.

---

### Decision · Program_Chairs · 2020-01-12

Accept